# Integrated Somatic and Germline Whole-Exome Sequencing Analysis in Women with Lung Cancer after a Previous Breast Cancer

**DOI:** 10.3390/cancers11040441

**Published:** 2019-03-28

**Authors:** Simona Coco, Silvia Bonfiglio, Davide Cittaro, Irene Vanni, Marco Mora, Carlo Genova, Maria Giovanna Dal Bello, Simona Boccardo, Angela Alama, Erika Rijavec, Claudio Sini, Valeria Rossella, Giulia Barletta, Federica Biello, Anna Truini, Cristina Bruzzo, Maurizio Gallo, Dejan Lazarevic, Alberto Ballestrero, Francesco Grossi

**Affiliations:** 1Lung Cancer Unit, IRCCS Ospedale Policlinico San Martino, 16132 Genoa, Italy; irenevanni85@yahoo.it (I.V.); carlo.genova1985@gmail.com (C.G.); mariagiovanna.dalbello@hsanmartino.it (M.G.D.B.); simona.boccardo@hsanmartino.it (S.B.); angela.alama@hsanmartino.it (A.A.); ery80x@yahoo.it (E.R.); audiosini@tiscali.it (C.S.); giulia.barletta@yahoo.it (G.B.); febiello@gmail.com (F.B.); anna.truini@gmail.com (A.T.); cristina.bruzzo@hsanmartino.it (C.B.); 2Centre for Translational Genomics and Bioinformatics, IRCCS San Raffaele Scientific Institute, 20132 Milan, Italy; bonfiglio.silvia@hsr.it (S.B.); cittaro.davide@hsr.it (D.C.); lazarevic.dejan@hsr.it (D.L.); 3Department of Internal Medicine and Medical Specialties (DIMI), University of Genoa, IRCCS Ospedale Policlinico San Martino, 16132 Genoa, Italy; maurizio.gallo@unige.it (M.G.); aballestrero@unige.it (A.B.); 4Department of Pathology, IRCCS Ospedale Policlinico San Martino, 16132 Genoa, Italy; marco.mora@hsanmartino.it; 5Swiss Stem Cell Biotech, 6830 Vacallo, Switzerland; valeria.rossella@stembiotech.ch; 6UOC Oncologia Medica Fondazione IRCCS Ca’ Granda Ospedale Maggiore Policlinico, 20122 Milan, Italy; francesco.grossi@policlinico.mi.it

**Keywords:** breast cancer, lung cancer, multiple cancer susceptibility, genetic predisposition, exome sequencing

## Abstract

Women treated for breast cancer (BC) are at risk of developing secondary tumors, such as lung cancer (LC). Since rare germline variants have been linked to multiple cancer development, we hypothesized that BC survivors might be prone to develop LC as a result of harboring rare variants. Sixty patients with LC with previous BC (the study population; SP) and 53 women with either BC or LC and no secondary cancer (control population; CP) were enrolled. Whole exome sequencing was performed in both tumors and unaffected tissues from 28/60 SP patients, and in germline DNA from 32/53 CP. Candidate genes were validated in the remaining individuals from both populations. We found two main mutational signature profiles: S1 (C>T) in all BCs and 16/28 LCs, and S2 (C>A) which is strongly associated with smoking, in 12/28 LCs. The burden test over rare germline variants in S1-LC vs CP identified 248 genes. Validation confirmed *GSN* as significantly associated with LC in never-smokers. In conclusion, our data suggest two signatures involved in LC onset in women with previous BC. One of these signatures is linked to smoking. Conversely, regardless of smoking habit, in a subgroup of BC survivors genetic susceptibility may contribute to LC risk.

## 1. Introduction

Breast cancer (BC) is one of the most common cancers, together with lung and colon, and it represents the most frequent malignancy in females worldwide [1]. Currently, the optimal management of early BC patients by integrated treatments, including radio-, chemo- and hormone therapies [1], have led to significant mortality reduction, with almost 90% of patients surviving at five years [2]. Despite the improvement in survival, approximately 10% of women with BC develop a second primary tumor, and lung cancer (LC) is one of the most frequently diagnosed tumors [3,4]. Several factors have been linked to LC co-occurrence in BC survivors, such as primary cancer treatment-related effects as well as exposure to carcinogens in tobacco products [5,6,7]. For decades, clinicians have hypothesized that radiotherapy could increase the risk of developing a second unrelated malignancy. Indeed, many evidences have shown a significant increase in the relative risk of developing LC after a BC in women receiving radiation therapy after a latency period of 5–20 years [8,9,10]. In particular, the risk increased in the ipsilateral lung and was more clearly related to post-mastectomy radiation [9]. A synergic effect of radiotherapy with smoking habit has also been reported [6,8]. In particular, smoking women who received post-mastectomy radiotherapy showed a 40-fold increase risk of an ipsilateral LC compared with never smokers who did not receive any adjuvant radio-treatment [11]. Although the exposure to carcinogens has been considered a risk factor of LC in BC survivors, substantial evidences argue that the genetic predisposition may also play a role in the development of multiple cancers [12]. Although several genetic variants have been associated with LC risk [13,14], a cancer susceptibility signature in the development of LC in BC survivors has not been elucidated yet. The advent of Next Generation Sequencing (NGS) has deeply revolutionized our knowledge of cancer molecular landscape [15,16,17]. The screening of the entire DNA coding region by whole exome sequencing (WES) represents the best option, compared to the expensive whole genome sequencing, in the detection of rare genetic variants involved in unknown genetic syndromes and cancer susceptibility [18,19].

In the present study we performed a WES analysis of both tumor (BC and LC) and germline DNA in a cohort of 28 patients that developed LC after BC (study population; SP) to identify both somatic and germline signatures clarifying the genetic susceptibility associated to LC development. To unravel any genetic predisposition, we applied a gene-based burden test over rare germline variants in the SP compared to a control population (CP), i.e., patients with a BC without any history of secondary cancers. Then, the candidate genes were validated in an independent cohort of 28 women with LC after a previous BC and compared to controls (10 women with BC and 11 women with LC but without any history of secondary cancers).

## 2. Results

### 2.1. Patients and Clinical Characteristics

Sixty women who developed LC after a previous BC (SP) and 42 patients with a primary BC without any secondary unrelated tumor (CP) were enrolled in the study. In addition, 11 women surgically resected for an adenocarcinoma of the lung, with no additional cancer history, were also recruited as control (CP-LC) for the Validation Set (VS). The clinical features and treatment information of SP and CP are summarized in Table 1 and Table 2, respectively. 

The median age for SP at BC diagnosis was 60 years (range: 37–76 years) and LC occurred nearly 8 years after BC. At BC diagnosis, 46.7% (28/60) and 13.3% (8/60) of the patients were current or former smokers, respectively, whereas three women quit smoking at the onset of LC. The most frequent BC histotype was invasive ductal carcinoma (86.6%), two of which were in situ. The majority of women (71.7%, 43/60) underwent a breast-conserving surgical approach, while 28.3% (17/60) had mastectomy. Every BC patient received at least one local and/or adjuvant systemic treatment (Table 2). Among the radio-exposed women, more than half (52.4%; 22/42) developed an ipsilateral LC. Approximately one third of women (23/60) of our cohort were high-risk patients, i.e. smoker at the time of radiotherapy, although only three underwent mastectomy before radiotherapy [11]. With regard to LC, the most common histotype was adenocarcinoma (83.6%) followed by squamous carcinoma (8.2%), small cell lung cancer (4.9%) and large cell carcinoma (3.3%); notably, one patient (#SP-60) developed two synchronous tumors of the lung: adenocarcinoma and squamous LCs, both contralateral to the primary BC.

### 2.2. Mutational Profile and Copy Number Alteration (CNA) of SP Breast Cancers

Breast cancer genomic DNA (gDNA) samples were isolated from formalin-fixed and paraffin-embedded (FFPE) tumor specimens from the 28 patients of the study population discovery set (SP-DS), who were enrolled in a wide period of time (1990–2014). Although the samples exhibited a heterogeneous degradation status (Appendix A), WES library preparation and sequencing were fully successful. However, somatic analysis revealed an unusual high mutational rate in 7/28 samples, with several hyper-mutated genes, suggesting false positive mutational calls probably linked to the formalin fixation. Therefore, we excluded these samples from the downstream analyses. Within the remaining 21 BC samples, we found a total of 420 mutated genes already described in cancer (see Appendix B). Each tumor showed at least one mutated gene (2–128), except sample #BC13 (Appendix A). Among the most frequently mutated genes in our cohort (Figure 1), some of them (*CDH1, KMT2C, GATA3, PIK3CA* and *TP53*) are known driver genes in breast cancer [20]. Other frequently mutated genes were also found in cBioPortal BC studies, although at lower frequencies than in our cohort (e.g., *BUB1B, MAGI2, MGA, PCNT, PDCD11, SPEN,* and *TRIOBP*). Notably, two triple negative BCs (ER−/PR−/HER2−) reported activating mutations in genes already described in this molecular subtype, such as *ATP8B2, CDC27, DHX37, FIN* and *SHROOM4* [21,22]. We then evaluated whether the 420 cancer genes carrying somatic mutations in our samples were also altered by the occurrence of Copy Number Alteration (CNA). Notably, some genes (*ATR, CDH1, KMT2C, GATA3, MGA, PIK3CA* and *TP53*) were also deregulated as a consequence of CNA (Appendix A; Figure 1). 

### 2.3. Mutational Profile and CNAs of SP Lung Cancers

All 28 LC samples were collected at the Pathology Unit of the IRCCS Ospedale Policlinico San Martino (Genoa, Italy). Similarly to the BC group, despite FFPE LC gDNA exhibited a heterogeneous fragmentation, WES library preparation and sequencing were successful for all samples (Appendix A). The pair comparison between LC and matching BC samples (*n* = 21) revealed no common alteration. We then looked for pathways enriched for genes carrying somatic mutations. We found a total of five pathways significantly enriched (false discovery rate FDR < 0.01) in two tumors (#BC29 and #LC7), but no shared with paired samples. All pathways (R-HSA-5083636, R-HSA-5083625, R-HSA-5083632, R-HSA-977068, R-HSA-3906995) are related to the O-linked protein glycosylation; their enrichment is mainly caused by mutations in the mucin gene family (*MUC12, MUC17, MUC3A, MUC4*, and *MUC5B*). 

Among the 28 LC samples, a total of 666 genes described in cancer was found mutated (see Appendix B), which were hit by at least one mutation (1–140), except sample #LC02 (Appendix A). The most frequently mutated gene in our cohort was *TP53* (42.8%; 12/28) (Figure 2), in agreement with data from the literature [20]. Other known LC driver genes were found mutated in our cohort. For example, 10/23 (43.5%) adenocarcinoma LCs harbored pathogenic mutations in *EGFR* (8/10 in exon 19 and 2/10 in exon 21). Across the variants of exon 19, beyond the canonical InDels (COSM6223, COSM6225 and COSM12678) we also found a deletion (p.Glu746_Arg748del) and rare nucleotide substitutions (COSM24267 and COSM51497) that were previously described [23]. In addition, two *KRAS* somatic mutations (Gly12Asp and Gly12Cys) were found in 2/23 adenocarcinoma LCs. Other cancer genes frequently altered in LC according to cBioPortal (*e.g., ZFHX4, RYR2, CSMD3, FAT3*, and *RP1L1*) were also found mutated at a high frequency in our cohort (Figure 2 and Appendix A). The integration of mutational and CNA analysis, as a further mechanism of gene deregulation, showed both gains (*EGFR*) and losses (*TP53*) in some driver genes (Appendix A). 

Since an increase in length of deletions across the genome has been described in some radiation-associated second malignancies [24], we investigated the potential radiotherapy related effect in our cohort of patients by regressing the extent of InDels (as found in our WES data) and chromosome aberrations (by Fraction of Genome Altered FGA) of LC samples, comparing radiation-exposed patients with unirradiated subjects. We did not find any significant association for either InDels (best *p*-value for logistic regression: 0.08 for deletions in the range [−5,0]) or FGA (logistic regression *p*-value = 0.844).

### 2.4. Microsatellite Instability (MSI) Analysis

Alteration of the DNA mismatch repair system has been associated to hereditary cancer syndromes [25]. To this end, we evaluated the MSI status by assessing five nearly monomorphic markers on both BC and LC samples from 27/28 SP-DS patients (in patient #SP-23 the amplification of 4/5 markers of the normal gDNA failed, preventing MSI assessment in both tumors). In addition, all markers of five BC samples, previously excluded from WES, failed to amplify as a probable consequence of highly fragmented gDNA (Appendix A). MSI was assessed by comparing BC and LC profiles with the matching normal sample (Appendix A). No MSI was found in BC or LC; these data were also confirmed on WES data as measured by MANTIS score (Appendix A). In one out of four patients, who showed a heterozygosity status within the monomorphic markers, an allelic imbalance at the NR21 was identified (Appendix A). In addition, we also found imbalances at the polymorphic loci in three BCs and in three LCs, although no loss of heterozygosity was shared between the tumor pairs from the same patient.

### 2.5. Mutational Signatures

In order to identify potential mutagenic processes underlying the development of LC in women with previous BC, we performed a mutational signature analysis. Two main mutational signatures were extracted from our cohort of 49 tumor samples (21 BC and 28 LC) [15]. All mutational signatures were compared to COSMIC signatures previously described in human cancers (Figure 3a,b; Appendix A) [15,26,27,28]. Signature 1 (S1), characterized predominantly by ‘C>T substitutions’, was found weakly similar to COSMIC S30 (cosine distance = 0.52). This signature was associated to all BC samples (21/21) as well as to a subgroup of LC (16/28). Conversely, Signature 2 (S2), dominated by ‘C>A transversions’, was exclusively found in LC (more specifically, 8/12 adenocarcinoma, 2/12 squamous and 2/12 small cell carcinoma) and showed a positive correlation with smoking (logistic regression *p*-value = 0.004; Table 3); specifically, 10 out of 12 patients were current/former smokers. Indeed, the S2 profile matched with COSMIC S4 (cosine distance = 0.35) which has been associated with smoking behavior. Since LC samples revealed two distinct mutational signatures, we next examined the clinical data of the two LC subgroups, as well as the potential effect linked to previous therapy for BC (Table 3). The S1-LC subgroup was mainly enriched in adenocarcinoma and the occurrence interval from primary BC was generally shorter (S1-LC 3.7 years, range: 0–11 versus S2-LC 7.7 years; range: 2–21). Overall, the therapies were not associated with the specific signature. Taken together, our data suggest that two different mutagenic processes drove the development of LC in our cohort. The presence of a tobacco-related signature hints that exposure to tobacco products played a predominant role in LC development in S2-LC subgroup of BC survivors. Conversely, the shared signature between BC and S1-LC group may underpin a genetic predisposition contributing to the development of both cancers.

### 2.6. WES Germline Analysis and Validation

In order to unravel a genetic contribution to the development of LC after BC in SP patients characterized by S1-LC signature (*n* = 16), we analyzed the differential distribution of rare germline variants (Allele Frequency in ExAC < 1%) in this group versus 32 out of 42 CP patients affected by BC only. Our gene-based burden test was performed on a subset of 4618 genes carrying more than one rare variant, identifying 248 candidate genes enriched with rare variants in SP versus control individuals (combined FDR < 0.05) (Appendix A). Interestingly, our panel included genes already described to be cancer predisposition genes (*FAS, ITK, KIF1B, PGR* and *MET*) or previously reported as playing important roles in cancer (*ABI1, ARID5B, DNMT3A, HEY1, KDM5C, NCOA1, NUP98,* and *SLC34A2*), or in DNA repair process (*FANCF*) [29,30,31,32,33,34]. We validated the genes found in the discovery cohort by a custom panel targeting all the exons of the 248 candidate genes. The library preparation was performed in 49 patients from VS (28 SP, 10 BC-CP, and 11 LC-CP) (Appendix A). Library preparation and sequencing were successful for all samples (Appendix A). The distribution of rare germline variants was initially tested in a population with the highest similarity to our initial cohort. Initially, we excluded 19 women with a history of smoking (former/current smokers) from the cases, assuming that the carcinogenic effect of tobacco may be a major determinant of LC development. Burden test, performed on 63/248 genes carrying more than one rare variant in this cohort, revealed that only *GSN* gene resulted significantly associated to LC in non-smoker patients (*p*-value = 1.06e−5, FDR = 6.67e−4, Appendix A). We identified three rare single nucleotide variants in four SP patients, two in DS and two in VS. Specifically, the missense variants rs766602997 (ExAC-MAF = 0.0033%) and rs116185403 (ExAC-MAF = 0.27%) with a predicted moderate impact, were carried by one (#SP-43) and two patients (#SP-27 and #SP-59) respectively. The third variant, (c.1927G>T; p.Glu643*) identified in #SP-03 patient, is a novel one and has a predicted loss-of-function. No gene was found significantly enriched in rare variants when smokers were included in the analysis, although 25 genes harbored at least two more rare variants in cases as compared to controls (Appendix A). Finally, to investigate additional biological processes that may contribute to the development of LC in patients with a previous BC, we aggregated the genes carrying at least one rare germline variant (allele frequency in ExAC < 1%) at the level of reactome pathways [35], and we then performed the burden test in SP patients with S1-LC signature versus 32/42 CP patients affected by BC only. We retrieved two pathways significantly enriched in cases versus controls (combined FDR < 0.05) (Appendix A).

## 3. Discussion

This study includes a population of 60 women surviving a BC, who developed a LC in an interval of approximately 7 years, in agreement with previous studies [8,9,10]. WES was performed in 28 patients including both tumors and normal tissue. No common mutation was shared between the two primary cancers, indicating their independent origin. Similarly, the pathway analysis did not disclose any common enrichment across paired samples but processes linked to mucin genes were highlighted in two unrelated tumor samples (a BC and a LC from two patients); their role in tumorigenesis remains questionable [36] as they are highly polymorphic [37] and many of them have been ruled out from the list of known cancer genes [38]. 

Analysis of somatic mutations revealed that the distribution of mutated genes in both BC and LC samples from SP-DS cohort is in line with reported studies for these tumors. Specifically, BCs harbored mutations in *PI3KCA, CDH1, TP53* and some other driver cancer genes already found in BC [20], although with weak differences in their frequencies. A possible explanation for this discrepancy might be due to the small number of samples analyzed as well as the unbalanced early stage BCs in our cohort; e.g., *TP53* is found more frequently mutated in metastatic patients than in early stage tumor [39]. The most frequent driver alterations in LC samples involved *TP53* and *EGFR* genes. We observed a higher frequency of *EGFR* mutations (36%) compared to published data (15–20%) in Caucasian patients [40]. However, our finding should not be surprising, since our cohort includes exclusively women and was enriched in never smokers with adenocarcinoma histology [41]. Deficiency of mismatch repair has also been described in the development of multiple tumors as well as in sporadic cancers, but at very low incidence (0.5–2%) [25,42]. Since data on this peculiar BC-LC population are still lacking, we investigated the MSI status. No significant result has been disclosed and data were also confirmed across all-coding regions by MANTIS score. Emerging evidences have demonstrated that each oncogenic process would leave a mutational signature in tumor DNA deriving from a combination of DNA repair deregulation and DNA-damaging agents e.g., smoking and radiations [28]. In this regard, Behjati and colleagues [24], investigating the genome of radio-induced tumors, identified radiation-related signatures irrespective of the tumor type. However, we did not find any significant pattern of alterations across the exome between radio-exposed *vs.* non-exposed women, hence we excluded any direct effect of radiotherapy on LC development in our cohort. Beyond radiation-related signatures, 30 distinct mutational profiles have been described from all cancers and some of them were linked to specific causes [15,26,27,28]. Analysis of mutational signatures in our cohort identified two main profiles characterized by the enrichment of distinct base substitutions: “C>T” (S1) and “C>A” (S2). Transition of C>T is one of the most common DNA modifications across the COSMIC signatures, and some of them have been linked to specific etiology, e.g., patient age [15]. However, our S1 was peculiar showing a slight similar pattern with COSMIC S30, described in a subgroup of BCs and of unknown etiology [43]. Conversely, the S2 profile clearly showed that tobacco smoking is linked to the emergence of LC in at least 12 samples; in this subset, the somatic signature resembles COSMIC S4, already linked to tobacco and confirmed in this study by regression with smoking habits [15,27,28]. According to these findings, we performed a gene-based burden test over rare germline variants restricted to a S1-LC of SP identifying 248 candidate genes enriched of rare variants compared to the CP. Among the genes significantly enriched in rare variants in SP-DS cohort, 14 were already known to be associated to cancer [32,33,34], suggesting that our cohort may at least cover known mechanisms of genetic predisposition to cancer. When we validated the panel on an independent cohort, a single gene (*GSN*) was found significantly enriched in rare variants in non-smoker cases vs. controls. The *GSN* gene (located on 9q33.2) encodes for a calcium-regulated protein mainly involved in the regulation of actin dynamics. The GSN protein has multifunctional roles inside the cell by coordinating several signal transduction pathways or acting as transcriptional regulator in both physiological and pathological conditions. In particular, GSN has been demonstrated to regulate apoptosis, proliferation as well as metastatic process in several cancers [44,45]. Evidences suggest that *GSN* may act as a tumor suppressor gene, indeed, several studies showed a GSN down-regulation in human cancers including breast carcinoma [44,46]. Notably, Wang and colleagues in 2014 reported that GSN may regulate the sensitivity to chemotherapy in patients with head-and-neck cancer [47], whereas more recently Zhao and colleagues have reported the GSN involvement in radio-resistant LC patients [48]. In support of this, Li and colleagues demonstrated a protective role of GSN against radiation-induced secondary injury in mice, by promoting the tissue regeneration [49]. Beyond the protein deregulation, genetic changes within its exon regions have been also linked to cancer predisposition such as breast and oral squamous cell carcinoma [46,50]. Taken together, these data suggest that patients harboring *GSN* gene defects may increase their susceptibility for developing a cancer as well as they would be more prone to the cancerous effects of the anticancer therapies. However, the validation on a larger cohort including smokers failed to score significant genes enriched in rare variants, possibly suggesting that the confounding effect of smoke might cover any genetic signal. 

Although our study shows that genetic background may influence the LC onset in BC patients providing new insights in LC etiology, our results should be read in light of some considerations. We are aware that the size of SP is relatively small, but it should be taken into account that the occurrence of secondary LC remains a rare event in patients with a previous BC. Indeed, when we checked public data (TCGA: Lung adenocarcinoma [LUAD] and Lung squamous cell carcinoma [LUSC] studies) we found only seven out of 411 female individuals with LC and previous BC and all of them were reported to be current/former smokers [51]. Contrarily to genome-wide association studies, WES-based studies are generally underpowered to evaluate the impact of single variants [52]; therefore, we chose to perform a burden test, aiming at the identification of genes which accumulate rare variants in our SP. We are aware that such tests do not weigh variants for their position in gene sequence nor the mutation type [53]. Therefore, we limited our analysis on variants predicted altering protein sequence. To enforce our results, we validated our candidate genes in an independent cohort of SP patients and CP. In particular, our CP included, over patients with BC without secondary malignancies, an additional group of women with lung adenocarcinoma and no history of multiple cancers, aiming to exclude any gene involvement in the LC carcinogenesis alone. This allowed us to select a list of genes, some of them already linked to cancer risk, among which *GSN* was confirmed to have a statistical significance. Although we analyzed only a small subgroup of BC survivors we found out rare variants in *GSN* in 16% (four out of 25) of the patients that are supposed to exhibit a genetic susceptibility to LC (S1-LC). In addition, the burden analysis at pathway level revealed that rare variants in the nicotinic acetylcholine receptor genes are enriched in S1-LC as compared with controls; interestingly, genetic variations in these receptors have been associated to lung cancer risk [54,55]. Finally, despite we excluded any direct effect of the radiation therapy on DNA, previous therapeutic treatments may also have contributed to LC onset, e.g., by lipid peroxidation and protein oxidation [56]. Therefore, we can assume that the development of LC in women with a previous BC might be the result of a combination of different factors, of which the genetic susceptibility would represent only a piece of the “omics” puzzle. 

## 4. Materials and Methods 

### 4.1. Patient Enrolment and Sample Collection

Between January 2006 and December 2017, 60 women (SP) with a LC diagnosis were admitted to the Lung Cancer Unit of the IRCCS Ospedale Policlinico San Martino (Genoa, Italy), reporting a medical history of previous BC in an interval ranging from few months to 31 years before LC (Appendix A). In 28/60 patients (DS), FFPE surgical specimens from both BC and LC were collected, together with normal tissue specimens, for the WES and MSI analysis (Appendix A). All tumor samples were checked by experienced pathologists for the presence of adequate tumor cell content (>50%). For the remaining 32/60 patients, tumor tissue from both tumors was unavailable for WES, therefore they were included in the VS and a normal tissue sample was provided for the germline analysis. In addition, two CPs including 42 women with primary BC (CP-BC) and 11 women with primary LC (CP-LC), without any history of multiple cancers, were enrolled and normal tissue samples were collected for each subject (Appendix A). Both CPs had a follow-up of at least 10 years (range: 10–16 years) from the cancer diagnosis (Appendix A). The present study was approved by the Local Ethics Committee (TrPo12.006) and conducted according to the provisions of the Declaration of Helsinki. For each patient included in the study a written informed consent was obtained.

### 4.2. Genomic DNA Extraction 

Genomic DNA (gDNA) from FFPE tumor and normal tissues was isolated using GeneRead DNA FFPE Kit or QIAamp® DNA Mini Kit (Qiagen, Hilden, Germany). The gDNA concentration was assessed by Qubit® 2.0 Fluorometer (Invitrogen, Carlsbad, CA, USA), whereas the FFPE gDNA degradation status was assessed by DNA Integrity Number (DIN) value on TapeStation 2200 (Agilent Technologies, Santa Clara, CA, USA). 

### 4.3. WES and Custom NGS Panel Library Preparation and Sequencing 

WES was performed using the Agilent SureSelect Human All Exon kit v6 (Agilent Technologies) as previously described [57]. Libraries were sequenced on an Illumina HiSeq 2500 platform, 2×100 bp (Illumina Inc., San Diego, CA, USA). A custom design targeting all exons of 248 genes (VS) was performed by Agilent SureDesign software (https://earray.chem.agilent.com/ suredesign/). Libraries were prepared using Agilent SureSelect XT HS kit (Agilent Technologies). Briefly, 200 ng of gDNA were fragmented on a Covaris E220 focused ultrasonicator (Covaris, Woburn, MA, USA) and libraries were prepared following the manufacturer’s protocol and sequenced on the NextSeq 500 platform, 2×100 bp (Illumina). Detailed information on the bioinformatics analysis is provided in the Appendix B. 

### 4.4. Burden Test

Burden tests were performed using the AssotesteR R package (https://cran.rproject.org/web/ packages/AssotesteR/index.html). In particular, the WSS test and C-alpha test were employed. The *p*-values of both tests were combined using Stouffer’s method and corrected for multiple tests [53,58]. 

### 4.5. MSI Analysis

MSI was evaluated by assessment of the status of 5 nearly monomorphic markers on both BC and LC of the 28 SP patients enrolled in the DS. MSI analysis was carried out by MSI Analysis System v. 1.2, (Promega, Madison, WI, USA) covering five quasi-monomorphic mononucleotide markers (NR21, BAT26, BAT25, NR24, and MONO 27) for MSI detection and two highly-polymorphic penta-nucleotide repeat markers (Penta C and Penta D) for specimen identification. A multiplex PCR assay was carried out according to the manufacturer’s protocol. Briefly, 8 ng of input gDNA was amplified according to manufacturer’s instruction. All PCR products were denatured in Hi-Di™ Formamide (ThermoFisher Scientific, Carlsbad, CA, USA) with Internal Lane Standard 600 (Promega) and analyzed on a 3130xl Genetic Analyzer using the GeneMarker analysis software (Softgenetics LLC, State College, PA, USA). Loss of heterozigosity was defined when the ratio of tumor alleles and normal alleles resulted <0.5 or >2.0 [59].

## 5. Conclusions

To the best of our knowledge, this study is the first attempt to probe the genetics underlying the development of consecutive primary breast and lung tumors in the same patient. Although the sample size of this study is relatively small, our data provide a comprehensive picture of this peculiar population. In particular, we confirm their independent origin; in addition, we also show that smoke, a well-established risk factor in the general LC population, also represents the main cause of LC in a subgroup of women with BC. However, we also report that in a subgroup of patients, supposedly no-smoking women, the genetic background may contribute to increase their susceptibility to LC development. In particular, we identified a list of candidate genes among which *GSN* was significantly associated to LC risk. More extensive well-powered and in-depth sequencing studies are required to confirm the link of *GSN* gene to the predisposition of LC in BC survivors.

## Figures and Tables

**Figure 1 cancers-11-00441-f001:**
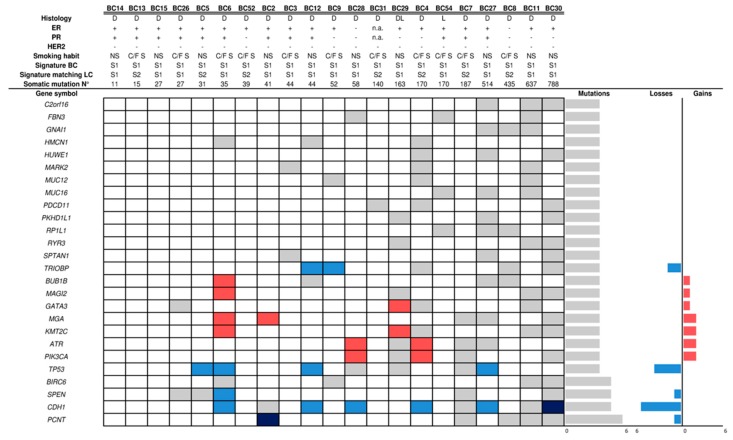
Mutated genes in breast cancer samples. The figure shows the mutational and copy number status of the 25 most frequently altered genes found in the cohort of BC samples obtained from SP patients. Grey, light red and light blue cells indicate the presence of a gene mutation, copy number gain and copy number loss, respectively. Dark red and dark blue cells indicate the concomitant presence of a gene mutation plus either gene copy number gain or loss, respectively. On the right of the picture, the number of mutations, and gene copy number losses and gains are shown. Abbreviations: BC: breast cancer; Histology: DL: ductal-lobular breast carcinoma, D: ductal breast carcinoma, L: lobular breast carcinoma; ER+: positive estrogen receptor (≥ 1%), PR+: positive progesterone receptor (≥ 1%); HER-2+: positive status (3+); Smoking Habit: NS: no smoker, C/F S: current/former smoker; Mutational Signature: S1: signature 1; S2: signature 2; N°: number; n.a.: not available.

**Figure 2 cancers-11-00441-f002:**
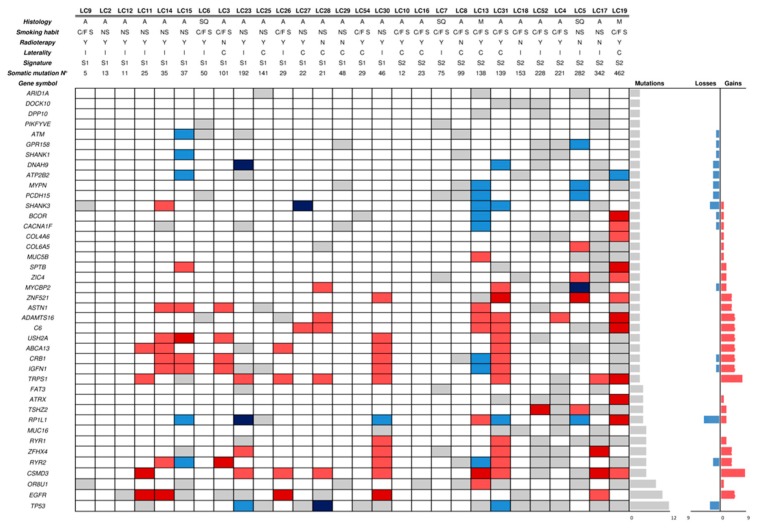
Mutated genes in lung cancer samples. The figure shows the mutational and copy number status of the 41 genes found most frequently altered in LC samples obtained from SP patients. Grey, light red and light blue cells represent the presence of a gene mutation, copy number gain and copy number loss, respectively. The dark red and dark blue cells represent the concomitant presence of a gene mutation plus either gene copy number gain or loss, respectively. On the right of the picture, the number of mutations, and gene copy number losses and gains are shown. Abbreviations: LC: lung cancer; A: adenocarcinoma; SC: squamous carcinoma; M: Microcitoma; NS: Never Smoker; C/F S: Current/Former Smoker; Y: radiotherapy; N: no-radioterapy; I: LC ipsilateral versus BC; C: LC contralateral versus BC; Mutational Signature: S1: signature 1; S2: signature 2; N°: number; n.a.: not available.

**Figure 3 cancers-11-00441-f003:**
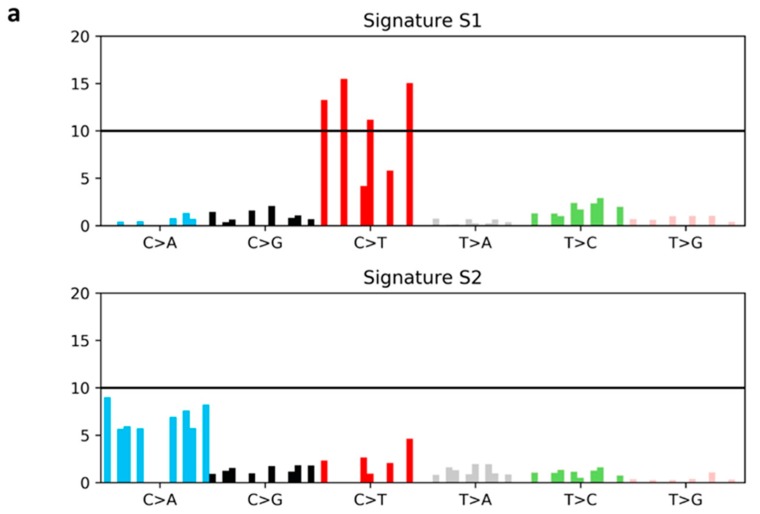
Mutational Signatures (S) of breast and lung cancer samples. (**a**) Mutational signatures extracted from WES data. Each signature is displayed according to the 96 pyrimidine substitutions in their trinucleotide context, grouped and colored by the 6 classes of base substitutions. X-axis reports the mutation types; Y-Axis shows the frequency of mutations attributed to a specific mutation type. (**b**) Contribution of each tumor type to the extracted mutational signatures. The BC samples are reported at the top whereas the LC samples were depicted at the bottom. BC samples not used in the analysis are masked in white. The dotted red line represents the threshold used to perform signature attribution. Abbreviations: BC: breast cancer; LC: lung cancer; A: adenine; C: cytosine; G: guanine; T: thymine.

**Table 1 cancers-11-00441-t001:** Clinical data of study and control populations.

Study Population						Control Population					
Breast Cancer n. 60			Lung Cancer n. 60			Breast Cancer n. 42			Lung Cancer n. 11		
Age at diagnosis	median	range	Age at diagnosis	median	range	Age at diagnosis	median	range	Age at diagnosis	median	range
Breast Cancer	60	37–76	Lung Cancer	68	48–88	Breast Cancer	60	41–81	Lung Cancer	59	37–81
Smoking habit	N.	%	Smoking habit	N.	%	Smoking habit	N.	%	Smoking habit	N.	**%**
Smoker	28	46.7	Smoker	25	41.7	Smoker	9	21.4	Smoker	3	27.3
Former smoker	8	13.3	Former smoker	11	18.3	Former smoker	2	4.8	Former smoker	1	9.1
Never smoker	23	38.3	Never smoker	23	38.3	Never smoker	27	64.3	Never smoker	6	54.5
n.a.	1	1.7	n.a.	1	1.7	n.a.	4	9.5	n.a.	1	9.1
BC histology	N.	%	LC histology	N.*	% *	BC histology	N.	%	LC histology	N.	**%**
Ductal Infiltrating	50	83.3	Adenocarcinoma	51	83.6	Ductal Infiltrating	33	78.6	Adenocarcinoma	11	100.0
Lobular	5	8.3	Squamous cell carcinoma	5	8.2	Lobular	7	16.7	Squamous cell carcinoma	0	-
Ductal/Lobular	1	1.7	Large cell	2	3.3	Ductal/Lobular	2	4.8	Large cell	0	-
In situ ductal	2	3.3	Small cell lung cancer	3	4.9	In situ ductal	0	-	Small cell lung cancer	0	-
n.a.	2	3.3	n.a.	0	-	n.a.	0	-	n.a.	0	-
Surgery type	N.	%	Surgery type	N.	%	Surgery type	N.	%	Surgery Type	N.	**%**
Quadrantectomy	35	58.3	Lobectomy	48	80.0	Quadrantectomy	12	28.6	Lobectomy	10	90.9
Lumpectomy	8	13.3	Segmentectomy	3	5.0	Lumpectomy	19	45.2	Segmentectomy	1	9.1
Mastectomy	17	28.3	Resection	2	3.3	Mastectomy	11	26.2	Resection	0	-
No surgery ^1^	0	-	No surgery ^1^	6	10.0	No surgery ^1^	0	-	No surgery ^1^	0	-
n.a.	0	-	n.a.	1	1.7	n.a.	0	-	n.a.	0	-
Disease stage	N.	%	Disease stage	N.	%	Disease stage	N.	%	Disease stage	N.	**%**
Non-metastatic	53	88.3	Non-metastatic	50	83.3	Non-metastatic	42	100.0	Non-metastatic	11	100.0
Metastatic	0	-	Metastatic	7	11.7	Metastatic	0	-	Metastatic	0	-
n.a.	7	11.7	n.a./n.a. ^2^	3	5.0	n.a.	0	-	n.a./n.a.^2^	0	-

* The sum and frequency of the lung cancer histotypes were calculated on 61 samples, since one patient developed two synchronous tumors of the lung (adenocarcinoma and squamous cell carcinoma). ^1^ Patient underwent a tumor biopsy. ^2^ Not applicable for small cell lung cancer histotype. Abbreviations: BC: Breast Cancer; LC: Lung Cancer; n.a.: not available.

**Table 2 cancers-11-00441-t002:** Breast cancer adjuvant therapies in study and control populations.

Study Population			Control Population		
Breast Cancer n. 60			Breast Cancer n. 42		
Adjuvant radiotherapy	N.	%	Adjuvant radiotherapy	N.	%
Yes	42	70.0	Yes	34	81.0
No	17	28.3	No	8	19.0
n.a.	1	1.7	n.a.	0	-
Adjuvant chemotherapy	N.	%	Adjuvant chemotherapy	N.	%
Yes	24	40.0	Yes	28	66.7
No	32	53.3	No	14	33.3
n.a.	4	6.7	n.a.	0	-
Hormone therapy	N.	%	Hormone therapy	N.	%
Yes	48	80.0	Yes	37	88.1
No	9	15.0	No	5	11.9
n.a.	3	5.0	n.a.	0	-

Abbreviations: n.a.: not available.

**Table 3 cancers-11-00441-t003:** Clinical data of study population patients according to mutational signature of lung cancer.

Clinical Data of Lung Cancer	S1		S2	
	N.	%	N.	%
Lung cancer	16	57.1%	12	42.9%
Age at diagnosis	67.5	56–75	68	53–78
LC occurrence after BC (years)	3.7	0–11	7.7	2–21
LC histotype	**N.**	%	**N.**	%
Adenocarcinoma	15	93.8%	8	66.7%
No adenocarcinoma	1	6.3%	4	33.3%
Smoking habit	**N.**	%	**N.**	%
Former/current smoker	5	31.3%	10	62.5%
Never smoker	11	68.8%	2	12.5%
Adjuvant radiotherapy for BC	**N.**	%	**N.**	%
Yes	13	81.3%	8	66.7%
No	3	18.8%	4	33.3%
Adjuvant chemotherapy for BC	**N.**	%	**N.**	%
Yes	6	37.5%	3	25.0%
No	10	62.5%	9	75.0%
Adjuvant hormone therapy for BC	**N.**	%	**N.**	%
Yes	15	93.8%	10	83.3%
No	1	6.3%	2	16.7%

Abbreviations: BC: breast cancer; LC: lung cancer; S: mutational signature.

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
