# Peer review of "Integrated Somatic and Germline Whole-Exome Sequencing Analysis in Women with Lung Cancer after a Previous Breast Cancer"

_cancers, 2019, doi:10.3390/cancers11040441_

Round 1
Reviewer 1 Report
This study is aimed to probe the genetics underlying the development of consecutive primary breast and lung tumors in the same patient. The results suggest that both smoke and genetic background contribute to lung cancer development in breast cancer survivors. Although the data provided a comprehensive description of the genetic characters of the normal and tumor tissues from this specific population, the data analysis was inadequate to extract the biological implications of the experimental data due to its limitation at the single-gene level.
Major concerns:
1. The pair comparison of genomic alterations between lung cancer and matching breast cancer samples is limited to analysis at the single-gene level. Since it is rare in nature that a phenotype is dictated by a single gene, additional analysis at the pathway and functional module levels should be conducted.
2. For the 4618 genes identified in SP patients characterized by S1-LC signatures to carry more than one rare germline variant, additional analysis should be conducted after aggregating genes at the levels of pathway and functional module, which could lead to identification of additional candidate genes for lung cancer predisposition.
3. It becomes clear that luminal and basal-like/triple negative breast cancer are different disease entities with distinct etiological factors. Detailed analysis of the status of ESR1, PGR, ERBB2 and Ki-67 proliferative index of breast cancer should be provided and their influence on genetic characters of subsequent lung cancer should be examined.
4. Multiple COSMIC mutation signatures have been linked to breast cancer. It is of great interest to provide an in-depth discussion on the potential underlying mechanism and implication of the finding that COSMIC S30 signature is prevalent in SP patients characterized by S1-LC signatures.
Minor points:
1. The procedure used to derive mutation signatures is missing in the Material and Method section.
2. Figure S1, define all the abbreviations in the figure legend.
Author Response
Reviewer 1
This study is aimed to probe the genetics underlying the development of consecutive primary breast and lung tumors in the same patient. The results suggest that both smoke and genetic background contribute to lung cancer development in breast cancer survivors. Although the data provided a comprehensive description of the genetic characters of the normal and tumor tissues from this specific population, the data analysis was inadequate to extract the biological implications of the experimental data due to its limitation at the single-gene level.
Dear reviewer#1 thank you for your suggestions; hereby our reply to the revision:
1 The pair comparison of genomic alterations between lung cancer and matching breast cancer samples is limited to analysis at the single-gene level. Since it is rare in nature that a phenotype is dictated by a single gene, additional analysis at the pathway and functional module levels should be conducted.
Author response:
We acknowledge the reviewer for these correct observations. In agreement with the reviewer’s suggestions, we performed enrichment analysis on mutated genes with Enrichr (the methods has been added to Appendix in the text (Line: 424). We then looked for pathways that are enriched by pairs of tumors in each patient. This analysis resulted in a limited number of processes related to protein O-glycosilation in two tumors (#BC29 and #LC7), but no shared between paired samples; such processes arose in our analysis due to the somatic mutations found in several mucin genes. Given that such genes are highly polymorphic and frequently mutated, we are persuaded that the pathways we found could be false positives; we conclude, then, that no common mechanism of tumorigenesis could be extracted from the analysis of somatic mutations. In attached the table listing the significant pathways extracted from 21 pairs of tumors (Table_Reviewer1#_Point_1). According to the results we included these data in both ‘Result’ and ‘Discussion’ sections as follows:
- "We then looked for pathways enriched for genes carrying somatic mutations. We found a total of five pathways significantly enriched (False Discovery Rate; FDR<0.01) in two tumors (#BC29 and #LC7), but no shared with paired samples. All pathways (R-HSA-5083636, R-HSA-5083625, R-HSA-5083632, R-HSA-977068, R-HSA-3906995) are related to the O-linked protein glycosilation; their enrichment is mainly caused by mutations in the mucin gene family (MUC12, MUC17, MUC3A, MUC4, and MUC5B)." (Lines: 127-132)
- "Similarly, the pathway analysis did not disclose any common enrichment across paired samples but processes linked to mucin genes were highlighted in two unrelated tumor samples (a BC and a LC from two patients); their role in tumorigenesis remains questionable [36] as they are highly polymorhic [37] and many of them have been ruled out from the list of known cancer genes [38]." (Lines: 227-231)
2. For the 4618 genes identified in SP patients characterized by S1-LC signatures to carry more than one rare germline variant, additional analysis should be conducted after aggregating genes at the levels of pathway and functional module, which could lead to identification of additional candidate genes for lung cancer predisposition.
Author response:
We acknowledge the reviewer for these correct observations. In agreement with the reviewer’s suggestions, we performed enrichment analysis on mutated genes with Enrichr (the methods has been added to Appendix in the text (Line: 424). We then looked for pathways that are enriched by pairs of tumors in each patient. This analysis resulted in a limited number of processes related to protein O-glycosilation in two tumors (#BC29 and #LC7), but no shared between paired samples; such processes arose in our analysis due to the somatic mutations found in several mucin genes. Given that such genes are highly polymorphic and frequently mutated, we are persuaded that the pathways we found could be false positives; we conclude, then, that no common mechanism of tumorigenesis could be extracted from the analysis of somatic mutations. In attached the table listing the significant pathways extracted from 21 pairs of tumors (Table_Reviewer1#_Point_1). According to the results we included these data in both ‘Result’ and ‘Discussion’ sections as follows:
- "Finally, to investigate additional biological processes that may contribute to the development of LC in patients with a previous BC, we aggregated the genes carrying at least one rare germline variant (Allele Frequency in ExAC <1%) at the level of Reactome pathways [35], and we then performed the burden test in SP patients with S1-LC signature versus 32/42 CP patients affected by BC only. We retrieved two pathways significantly enriched in cases versus controls (combined FDR <0.05) (Supplementary Table S11)." (Lines:217-222 ).
- "In addition, the burden analysis at pathway level revealed that rare variants in the nicotinic acetylcholine receptor genes are enriched in S1-LC as compared with controls; interestingly, genetic variations in these receptors have been associated to lung cancer risk [54,55]." (Lines: 304-307).
3. It becomes clear that luminal and basal-like/triple negative breast cancer are different disease entities with distinct etiological factors. Detailed analysis of the status of ESR1, PGR, ERBB2 and Ki-67 proliferative index of breast cancer should be provided and their influence on genetic characters of subsequent lung cancer should be examined.
Author response:
We acknowledge the reviewer for these right observations. Indeed, epidemiological studies suggest that gender can be an independent risk factor for lung cancer (Brinton et al., Cancer Epidemiol Biomarkers Prev. 2011). Emerging evidences have also demonstrated that the interplay between estrogen and intra-tumoral estradiol can influence LC tumorigenesis (Marquez-Garban et al., Steroids. 2011). Therefore, to better clarify the hormone role in the lung cancer risk in our cohort of women with a previous breast cancer, we have investigated the hormone receptor status (Estrogen and progesterone expression) together with ERBB2 and Ki67 in our cohort of women with double cancers with respect to two independent controls (females with a breast cancer or a lung cancer and no secondary neoplasia). Unfortunately, the analysis of the data did not show any significant findings. All these data are reported in a further paper currently under revision. We hope to be able to share these data with you and the scientific community as soon as possible. We also agree with the reviewer that triple negative breast cancer is a peculiar entity. In our cohort two out of 28 women are triple negative cancer. Accordingly to the reviewer’s suggestion we revised the exome data in this subgroup of tumors identifying activating mutations in genes (e.g. ATP8B2, CDC27, DHX37, FIN, SHROOM4) already described in triple negative breast cancers. This finding has been added in the main manuscript as follows: "Notably, two triple negative BCs (ER-/PR-/HER2-) reported activating mutations in genes already described in this molecular subtype, such as ATP8B2, CDC27, DHX37, FIN and SHROOM4 [21,22]." (Lines: 116-118).
4. Multiple COSMIC mutation signatures have been linked to breast cancer. It is of great interest to provide an in-depth discussion on the potential underlying mechanism and implication of the finding that COSMIC S30 signature is prevalent in SP patients characterized by S1-LC signatures.
Author response:
We acknowledge the reviewer for this relevant question. To date, mathematical algorithms have yielded numerous mutational signatures, based on an analysis of tens of thousands of exomes and genomes. At time of writing, 30 COSMIC signatures (https://cancer.sanger.ac.uk/cosmic/signatures) have been described across the spectrum of human cancer types and about half have been linked to a specific cause, such as exogenous mutagen exposition (e.g. smoking, ultraviolet light and alkylating agents) as well as genetic modifications, both germinal (BRCA1-2, APOBEC3A and APOBEC3B) and somatic (POLE). Our signature S1, dominated mainly by C>T, reported a similarity with the previous COSMIC S30 described in a small subset of breast cancer, but its etiology has not been discovered yet. Unfortunately, our analysis did not identify any specific genes or pathways potentially underlying its mechanism. A possible explanation for our non-informative results may be associated to the intrinsic limitations of the technology; indeed, despite we applied a whole exome approach sequences all coding exons (approximately 180.000), it represents less than 2% of the whole genome. Therefore, we cannot exclude that alterations in non-coding regions, involving for instance microRNAs, as well as further epigenetic mechanisms or expression pathways, might be involved in this signature. Indeed, it is known that microRNAs (Landau DA et al., Semin Oncol. 2011; Arjumand W et al., Methods Mol Biol. 2018) and gene methylation (Hawkins NJ et al., Mod Pathol. 2009) can increase the rate of DNA mutations, acting as a mutagenic factor. Another potential limitation may be the relatively small cohort of patients that may have prevented us to draw significant conclusions. However, our study is the first deep sequencing analysis that shows a rare mutational signature shared by women with a susceptibility to secondary cancers. We believe that our observations can provide a starting point for future studies aiming at better understanding any potential mechanisms in the COSMIC S30 etiology and association to lung cancer susceptibility.
Minor points:
1. The procedure used to derive mutation signatures is missing in the Material and Method section.
Author response:
We apologize for not having shared comprehensively this relevant information. Therefore, a detailed procedure on the signature extraction has been added at the end of the manuscript in the ‘Appendix’ section (Lines: 418-421).
2. Figure S1, define all the abbreviations in the figure legend.
Author response:
We apologize for not having correctly reported the abbreviations. A detailed list of all abbreviations has been added at the end of the figure and table legends.

Reviewer 2 Report
Manuscript by Coco et al is well designed and written. Authors found two main signature associated with lung cancer onset in women treated for breast cancer. This study includes 60 women surviving a breast cancer and later on having lung cancer.
Author Response
Reviewer 2
Manuscript by Coco et al is well designed and written. Authors found two main signature associated with lung cancer onset in women treated for breast cancer. This study includes 60 women surviving a breast cancer and later on having lung cancer.
Dear reviewer#2 thank you for your suggestions; hereby our reply to the revision:
We thank the reviewer for having expressed a favorable opinion on our study. We hope that our findings can provide useful information on the susceptibility to cancer and a starting point for future studies aimed at selecting a subgroup of breast cancers with a higher risk of developing lung cancer.

Reviewer 3 Report
In this manuscript, authors performed whole exome sequencing in both tumors and unaffected tissues from 28 SP patients with Lung cancer subsequent to Brest cancer and found two main mutational signature profiles: S1 (C>T) in all patients with Breast cancer and 16/28 in patients with Lung cancer, and S2 (C>A) in 12/28 in patients with Lung cancer, which is strongly associated with smoking. GSN gene also significantly associated to Lung cancer in no-smoker 4 patients with lung cancer. However, there seems to be some major limitations.
Major revisions:
Firstly, the cohort of patients with breast cancer is genetically heterogeneous which includes hormone receptor positive, HER2 positive and triple negative breast cancer. Subsequently lung cancers are also heterogenous including adenocarcinoma, squamous cell carcinoma, and small cell carcinoma. It is difficult to evaluate the same mechanism of carcinogenesis using these patients’ population.
C>T signature may well only represent sequence artifacts in FFPE samples which caused by deamination of cytosines.
The sequence context of the mutation or the transcriptional strand on which it occurs, can be incorporated into the set of features by which a mutational signature is strictly defined (Nature. 2013 August 22; 500(7463): 415–421). In this study, exome sequencing results should be followed to this definition of mutational signature in human cancer.
In this cohort, more than half of cases have smoking history, which represents that the number of non-smokers is too small to draw definite results.
Genetic susceptibility of GSN gene alteration should be evaluated by comparison with frequency pf snp in the general population.
It is necessary to verify the pathogenicity for each mutation in silico (such as CADD score, FATMM, SIFT and PolyPhen).
Such as EGFR, KRAS mutational status and ALK fusion on patients with Lung cancer gene may be needed.
Discussion concerning the mechanism in which GSN affect the carsonogenesis of patients with lung cancer is insufficient.
Minor revisions:
Table1: Total number of LC histology in the SP group is 61 cases.
Author Response
Reviewer 3
In this manuscript, authors performed whole exome sequencing in both tumors and unaffected tissues from 28 SP patients with Lung cancer subsequent to Brest cancer and found two main mutational signature profiles: S1 (C>T) in all patients with Breast cancer and 16/28 in patients with Lung cancer, and S2 (C>A) in 12/28 in patients with Lung cancer, which is strongly associated with smoking. GSN gene also significantly associated to Lung cancer in no-smoker 4 patients with lung cancer. However, there seems to be some major limitations.
Dear reviewer#3 thank you for your suggestions; hereby our reply to the revision:
1. Firstly, the cohort of patients with breast cancer is genetically heterogeneous which includes hormone receptor positive, HER2 positive and triple negative breast cancer. Subsequently lung cancers are also heterogenous including adenocarcinoma, squamous cell carcinoma, and small cell carcinoma. It is difficult to evaluate the same mechanism of carcinogenesis using these patients’ population.
Author response:
We acknowledge the reviewer for these correct observations. We are aware of the heterogeneity means as phenotype (hormone status in the breast cancers) as well as different histology across the lung cancers in our cohort of patients. With regard to hormone status, epidemiological studies suggest that gender can be an independent risk factor for lung cancer (Brinton et al., Cancer Epidemiol Biomarkers Prev. 2011). Emerging evidences have also demonstrated that the interplay between estrogen and intra-tumoral estradiol can influence lung cancer tumorigenesis (Marquez-Garban et al., Steroids. 2011). Therefore, as pointed out by the reviewer, and to better clarify the hormone role in the lung cancer risk in our cohort of women with a previous breast cancer, we have investigated the hormone status (Estrogen and progesterone expression) in our cohort of women with double cancers with respect to two independent controls (females with a breast cancer or a lung cancer and no secondary neoplasia). The analysis did not show any significant findings leading to hypothesize that this factor should not represent a bias in our analysis. All these data are reported in a further paper currently under revision; we hope to be able to share these data with you and the scientific community as soon as possible. As for the different lung cancer histotypes included in the study, we agree with the reviewer that it may be a potential confounding factor in the analysis. However, the extraction of mutational signatures across the lung tumors showed two main profiles: the first (S1) is enriched in adenocarcinomas (15/16 of lung cancer), whereas the remaining histotypes were gathered together in a signature clearly associated with the tobacco smoking. This finding is not surprising as it has indeed been demonstrated that both squamous cell carcinoma and microcitoma are more frequently observed with increasing exposure to tobacco smoke (Pesch B et al., Int. J of Cancer 2012). Therefore, we can assume that despite the heterogeneity of the patient cohort, our data provide a comprehensive picture of this peculiar population.
2. C>T signature may well only represent sequence artifacts in FFPE samples which caused by deamination of cytosines.
Author response:
We acknowledge the reviewer for this correct observation. We agree on the fact that formalin-fixation may damage DNA in different ways and that the "C>T" and "G>A" substitutions are the most common Formalin-Fixed Paraffin-Embedded (FFPE) artefacts. As for this observation, in a previous study (Bonfiglio et al., BMC Cancer 2016) we included in the gDNA isolation procedure a treatment with uracil-DNA glycosylase (UDG), aimed at removing the cytosine deamination artefacts and minimizing the risk of false variant calls. We hence investigated the prevalence of known FFPE artefacts (C>T and G>A substitutions) in FFPE samples treated with UDG, comparing the exome data with their matching fresh frozen tumor samples. The genotype concordance rate between each matched FFPE pairs, computed for C>T and G>A substitutions, were found in line with other transition rates. In the same way, we added the UDG treatment in the DNA isolation from the tumor and normal tissue samples in our patient cohort. However, despite this trick, in 7 out of 28 breast cancers the somatic analysis revealed an unusual high mutational rate, suggesting false positive mutational calls probably due to formalin fixation. Interestingly, 3/7 breast tumor FFPE blocks were very old (before 2000). This data is not surprising, indeed it is known that recent FFPE blocks are preferable to older one for DNA mutational analysis and possible reasons may be linked to environmental factors related to the storage facility as well as factors unique to historical tissue processing (Prentice L.M. et al., Plos One 2018). Anyway, we excluded these samples from the downstream analyses. In addition, to further minimize the base call errors linked to sample treatments as well as sequencing errors, we applied a stringent pipeline which allowed to exclude likely false positive variants affected by strand bias, as described in Appendix A - Somatic analysis. Nonetheless, we are aware that despite our tricks, we cannot completely exclude false positive calls. However, when we checked out the list of mutated genes in the breast and lung cancer samples within catalogues of somatic mutations from cancer genomes (IntoGen, Cancer Gene Census, and DriverDBv2), the majority of them were already described in cancer. Last but not least, it is important to note that more than half of the COSMIC signatures reported a pattern of C>T transitions, leading to hypothesize that this modification may represent one of the most common mutational processes addressing the development of cancer.
3. The sequence context of the mutation or the transcriptional strand on which it occurs, can be incorporated into the set of features by which a mutational signature is strictly defined (Nature. 2013 August 22; 500(7463): 415–421). In this study, exome sequencing results should be followed to this definition of mutational signature in human cancer.
Author response:
We acknowledge the reviewer for these correct observations. We apologize if the text was not clear enough, indeed we applied the same strategy suggested by the reviewer to extract signatures of somatic mutations. We have added details about our procedure in the Appendix section as follows: "Counts of all 96 possible pyrimidine to purine substitution in their trinucleotide context were used to build a frequency matrix. Non-negative Matrix Factorization was then applied; choice of the number of factors was based on the profile of the reconstruction error (divergence) curve." (Lines: 418-421).
4. In this cohort, more than half of cases have smoking history, which represents that the number of non-smokers is too small to draw definite results.
Author response:
We acknowledge the reviewer for this correct observation. We are aware that the size of our study population (women with a lung cancer after a previous breast cancer) is relatively small and the number of non-smoker women decrease further in this cohort. However, it should be taken into account that the occurrence of secondary LC remains a rare event in patients with a previous BC. Indeed, when we checked public data (TCGA: LUAD and LUSC studies) we found only 7 out of 411 female individuals with LC and previous BC and all of them were reported to be current/former smokers (Grossman et al., N Engl J Med. 2016). In addition, it is clearly demonstrated that the smoking is the biggest preventable cause of lung cancer. However, we found that approximately half of patients were non-smoker, and their lung cancer showed a peculiar mutational signature shared with the breast cancer pair from the same patient. These data led to hypothesize that in a subgroup of women with breast cancer, the genetic background may contribute to increase their lung cancer risk. Indeed, the germinal analysis (restricted on this patient subgroup) identified more than 200 candidate genes enriched of rare variants. We are aware that well-powered and in-depth sequencing studies are needed to confirm our findings however our observations may provide a starting point for future studies aiming at better understanding the cancer genetic predisposition.
5. Genetic susceptibility of GSN gene alteration should be evaluated by comparison with frequency pf snp in the general population.
Author response:
We acknowledge the reviewer for this advice. We checked ExAC database and we found that GSN gene typically carries less variant than expected in the general population, although the deviation from the expected number is not significant (Z score is 0.78 and 1.75 for synonymous and missense variants, respectively).
6. It is necessary to verify the pathogenicity for each mutation in silico (such as CADD score, FATMM, SIFT and PolyPhen).
Author response:
We acknowledge the reviewer for this correct observation. The predicted impact on protein function had been previously computed in dbNSFP using different predictors (Polyphen2, SIFT, FATHMM, CADD) (see Appendix 1 in the text), so we extracted the relevant fields from the original vcf files and we added them as further 5 columns in Supplementary Table S3 and S5.
7. Such as EGFR, KRAS mutational status and ALK fusion on patients with Lung cancer gene may be needed.
Author response: We acknowledge the reviewer for this advice. The mutational status of both EGFR and KRAS as well as the ALK rearrangement have been added, when available, in the Supplementary Table S1.
8. Discussion concerning the mechanism in which GSN affect the carsonogenesis of patients with lung cancer is insufficient.
Author response:
We acknowledge the reviewer for this correct observation. As correctly suggested, we deeply discussed the role of the Gelsolin (GSN) as follows: "The GSN gene (located on 9q33.2) encodes for a calcium-regulated protein mainly involved in the regulation of actin dynamics. The GSN protein has multifunctional roles inside the cell by coordinating several signal transduction pathways or acting as transcriptional regulator in both physiological and pathological conditions. In particular, GSN has been demonstrated to regulate apoptosis, proliferation as well as metastatic process in several cancers [44,45]. Evidences suggest as GSN may act as a tumor suppressor gene, indeed, several studies showed a GSN down-regulation in human cancers including breast carcinoma [44,46]. Notably, Wang and colleagues in 2014 reported as GSN may regulate the sensitivity to chemotherapy in patients with head-and-neck cancer [47], whereas more recently Zhao and colleagues have reported the GSN involvement in radio-resistant LC patients [48]. In support of this, Li and colleagues demonstrated a protective role of GSN against radiation-induced secondary injury in mice, by promoting the tissue regeneration [49]. Beyond the protein deregulation, genetic changes within its exon regions have been also linked to cancer predisposition such as breast and oral squamous cell carcinoma [46,50]. Taken together these data suggest that patients harboring GSN gene defects may increase their susceptibility for developing a cancer as well as they would be more prone to the cancerous effects of the anticancer therapies." (Lines: 268-282).
Minor revisions
Table1: Total number of LC histology in the SP group is 61 cases.
Author response:
We apologize for this confounding information. Indeed, as the reviewer pointed out, the study population includes 60 patients, whereas the table reports 61 tumors. However, one out of 60 patients developed during her life two synchrony lung cancers, adenocarcinoma and squamous lung tumor, both contralateral to the primary breast cancer. Therefore, in order to improve the readability of these data we added an asterisk in the sum and the frequency of lung cancer histotypes. This information was also added in the table legend (Lines: 86-87).

Round 2
Reviewer 1 Report
Revision is acceptable for publication
Reviewer 3 Report
Reviewer's comments:
In this revised manuscript, issues pointed out have been improved. However, there seems to be minor issues to be declared.
Authors reply that all data concerning breast cancer phenotype are reported in a further paper currently under revision. It may correspond to double publication, because double publication can be started with double (or multiple) submission (Korean J Fam Med. 2012 Mar; 33(2): 69.). However, unless the heterogeneity of the cohort is resolved, it is difficult to draw definite results.
Authors’ responses:
Reviewer 3#
In this revised manuscript, issues pointed out have been improved. However, there seems to be minor issues to be declared.
Authors reply that all data concerning breast cancer phenotype are reported in a further paper currently under revision. It may correspond to double publication, because double publication can be started with double (or multiple) submission (Korean J Fam Med. 2012 Mar; 33(2): 69.). However, unless the heterogeneity of the cohort is resolved, it is difficult to draw definite results.
Dear reviewer#3 thank you for your suggestion; hereby our reply to the revision:
Dear Reviewer, we apologize for the misunderstanding about the data on breast cancer phenotype. The mentioned paper “under revision” involved only a subgroup of the patients (37/60) and a larger control group, including 42 patients with lung cancer and no previous tumor. The study examined the hormone status (PR, ER and HER2) in both cancers from women with double malignancies and controls. The hormone status in our population was in agreement with the general population i.e. women with a primary breast cancer only (Clark GM, et al., J Clin Oncol. 1984). In addition, only 10% of the triple negative breast cancers were in accordance with data from the literature (Foulkes WD, et al., N Engl J 2010). Since emerging evidences have demonstrated that the interplay between estrogen and intra-tumor estradiol can influence the lung cancer tumorigenesis (Marquez-Garban et al., Steroids. 2011), we also examined the hormone status in the paired lung cancer from the same breast cancer patients and then compared results with the control lung cancer population. However, the majority of lung cancers from patients with double malignancies, as well as the matched lung cancer controls were negative for both hormone receptors and HER2. The above mentioned information is, however, confidential. Finally, we have also investigated the protein expression of a small panel of markers involved in cancer. All markers were unrelated to the data from the Manuscript ID: cancers-452240. Considering the different marker sources (whole exome data vs protein data) and the different results we obtained from these two studies, we believe that the two manuscripts are unrelated and do not correspond to double publication.
Regarding your previous comment (point 1) we did not find any specific correlation between the genome data and the hormone expression, and only a few number of genes were found in the subset of triple negative patients (2/28 women; See lines 116-118 of the revised manuscript). Therefore, we believe that despite the heterogeneity of the patient cohort, our data provide a reliable picture of this peculiar population.
Reviewer's feedback:
My evaluation is that it is
acceptable in this form.